# Research on Multi-Fault Diagnosis Method Based on Time Domain Features of Vibration Signals

**DOI:** 10.3390/s22218164

**Published:** 2022-10-25

**Authors:** Chao Wang, Zhangming Peng, Rong Liu, Chang Chen

**Affiliations:** School of Mechanical Engineering, Hangzhou Dianzi University, Hangzhou 310018, China

**Keywords:** engine, fault diagnosis, GRU, Pearson correlation coefficient

## Abstract

The normal operation of the engine is of great importance for the safety of life and property, so we need to monitor and analyze the state of the engine. Most of the existing methods only diagnose the type of engine fault without further analysis of the severity of the engine fault. Additionally, the features used for fault diagnosis are not selected according to faults and do not necessarily contain more fault information. In the paper, we propose using Pearson correlation coefficients in combination with faults selects sensors and the corresponding features, and then single-fault diagnosis combined with GRU (gating recurrent unit) is performed by using the selected sensors and features. Since multi-fault diagnosis is more difficult than single-fault diagnosis, more state information is required. Therefore, the multi-fault diagnosis will directly extract the time domain features screened above from all vibration signals, stack them and send them to GRU for multi-fault diagnosis. From the experimental results we can conclude that the feature selection method combining Pearson correlation coefficient and fault state can extract effective features to diagnose the fault type and its severity. Finally, the influence factors of the model are analyzed through comparative experiments, and the results show the effectiveness of the method and the selected model parameters.

## 1. Introduction

The engine is the core power source of important equipment. It is particularly important for the normal operation of the equipment to detect and deal with the faults of the starting machine in time, which can greatly increase the safety of life and reduce the property loss.

Fault diagnosis methods based on artificial intelligence do not rely on the expert knowledge and experience and can achieve good results, so the paper uses artificial intelligence methods for fault diagnosis. As monitoring data are mostly time series signals, extracting the time series information from them is especially important for fault diagnosis, RNN (recurrent neural network) is widely used in fault diagnosis because of its excellent ability of time domain information mining. Although RNN is more effective in processing time series data, it faces the problem of vanishing gradient, which is alleviated by GRU to some extent. Gao [1] used GRU to diagnose the collected sample set and compared it with BP (back propagation) network, and the results showed that GRU had higher accuracy. Shi [2] used bidirectional GRU for fault diagnosis and showed high diagnostic accuracy. In Zhang [3], CNN (convolutional neural network) and GRU are simultaneously used for feature extraction to complete fault diagnosis. GRU has high accuracy, fewer parameters, and high computational efficiency, so GRU is selected as the selection network in this paper.

Extraction of effective features can improve fault diagnosis accuracy, and various methods are used to extract effective features. For example, Zhou [4] obtained the product of the square of the instantaneous amplitude and the instantaneous frequency of the vibration by using the Teager operator, effectively extracted the instantaneous energy of the cylinder vibration signal, and used it as a feature for the fault diagnosis of the cylinder. When selecting sensor signals for fault diagnosis, sensors are often selected according to the physical location of the research target. However, a sensor that is consistent with the physical location of the research target does not mean that it carries the most fault-related information. At the same time, many methods use VMD (variational mode decomposition), wavelet transform, and other algorithms to deeply mine the frequency domain information of the sensor signal after selecting the sensor signal and extracting the frequency domain information for fault diagnosis. For example, Wang [5] uses wavelet transform to extract the information of instantaneous speed signals in normal state and fault state and compares them to obtain effective fault information. Jiang [6] also obtained the effective frequency domain information of cylinder misfire fault by using wavelet transform. Bi [7], Zhang [8], and Qiao [9] use VMD to extract information from multiple modes and use it for fault diagnosis. For example, Yang [10] used correlation analysis to obtain characteristic correlation coefficients in the time domain, frequency domain, and time–frequency domain under different working conditions to obtain sensitivity features and used the normal state as the benchmark for fault diagnosis. Xu [11,12] used wavelet packet transform to extract the energy entropy of each node and Hilbert transform to extract the marginal spectrum energy entropy and sent it as a feature to SVM (support vector machine) for fault diagnosis. However, the performance of algorithms such as VMD and wavelet transform depends on the selected parameters to a large extent. At the same time, there is no complete theory about the selection of these parameters to select reasonable parameters for fault diagnosis. Therefore, it is difficult to extract frequency information for fault diagnosis. The time domain feature is used for fault diagnosis in this paper. To select the signals and features carrying more fault information, this paper proposes to use the Pearson correlation coefficient combined with fault state to extract time domain features for fault diagnosis.

The fault type and severity diagnosis of the engine are particularly important for taking reasonable maintenance measures. However, most of the existing methods only diagnose the fault type without further classification of the fault severity. These classification methods are not conducive to taking reasonable maintenance measures. Some scholars have taken some methods to analyze the severity of faults in the fields of bearings and gears. Xiao [12] uses wavelet transform and curve fitting to obtain warning value and alarm value and judges the fault degree based on the fault indication value obtained by fitting. This method is highly dependent on the selection of wavelet function and needs to determine the threshold manually, which is highly dependent on expert knowledge and experience. Pan [13,14] first performed feature selection and dimensionality reduction on the data. Then, he fed the selected features into a monotonic decision tree for fault severity diagnosis. The accuracy of his method reached 93.58%. JHA [14] completed the fault severity diagnosis of the rolling bearing in two stages by using a multi-classification support vector machine. Machine learning methods such as support vector machines can only extract shallow features. Therefore, when the amount of data is large, the generalization ability of machine learning methods is poor, and the data information cannot be mined well. Almounajjed [15] completed the fault severity diagnosis of the stator gate short circuit of the induction motor by using the mathematical model. Hang [16] completed the detection of fault severity of high resistance wiring of the synchronous motor by establishing a mathematical model. Taha [17] combined dissolved gas analysis and neural pattern recognition to complete the fault severity diagnosis of the power transformer. The accuracy of his method reached 93.58%. The model-based fault severity diagnosis method is difficult to implement in many cases because it requires professional knowledge and rich experience to establish the model, and the generalization ability of the model-based method is generally weak. Yang [18] completed the fault severity diagnosis of the rolling bearing by integrated learning, with a training accuracy of 98.57%, verification accuracy of 100%, and test accuracy of 96.51%. Gai [19] completed the fault severity diagnosis of rolling bearings using DBN (deep belief network), and the average detection accuracy reached 96%. The fault severity analysis of the above method is aimed at fewer fault severity types, and its ability to solve complex problems is not high. Sun [20] completed the diagnosis of six fault types and their corresponding four fault severity levels by using a multi-head attention network combined with deep learning. However, the existing method requires environmental and control variables. These data requirements cannot be realized in many practical situations. Dibaj [21] completed the fault severity diagnosis of bearings and gears by using VMD and CNN. However, this method needs to set the parameters of VMD and select the threshold. The reasonableness or otherwise of the parameters has an impact on the fault diagnosis results. The fault diagnosis results can only be obtained by further analyzing the results obtained by the network.

Most of the existing fault severity diagnosis methods require more demanding raw data and final results require further analysis of the model diagnosis results or the setting of thresholds based on experience. The selection of data processing methods and their parameters greatly depends on experience and professional theoretical knowledge. In order to select features containing more fault information, a feature selection method combining the Pearson correlation coefficient and faults is proposed in the paper to select features for fault severity diagnosis. First, sensors containing more fault information and the corresponding time domain features are selected twice using Pearson correlation coefficients [22]. Then, the time domain features are sent to the GRU as input for fault diagnosis. Since multi-fault diagnosis is more difficult and requires more state information, we select the time domain features selected above from all vibration signals and concatenate them as input to GRU for multi-fault fault diagnosis.

## 2. Theory and Methods

### 2.1. Theory

Most of the monitoring sensor signals of the engine are time series data, and the RNN is more effective in processing the time series data. However, RNNs suffer from the problem of gradient disappearance, which is alleviated by the proposed LSTM and GRU. Compared with LSTM, GRU has fewer parameters, high computational efficiency, and can achieve almost the same accuracy as LSTM.

#### GRU

Since monitoring data of engines are mostly time series data, the processing ability of the [23] time series data of the RNN network is stronger. The conventional RNN has problems such as short-term memory and gradient disappearance, and GRU overcomes the shortcomings of RNN by adding a gate structure. Therefore, GRU is chosen in the paper as the network of choice for fault type and its severity diagnosis. GRU unfolding structure is shown in Figure 1.

Figure 1 shows GRU unfolding structure [24]. The current hidden state $h_{t}$ is associated with the hidden state $h_{t-1}$ at the previous time and the current input $x_{t}$. $h_{t-1}$ contains the information at time T − 1, which is the historical information used when obtaining the current state. The reset gate $r_{t}$ is used to control the transition from the historical information $h_{t-1}$ to the hidden candidate state ${\tilde{h}}_{t}$ of the regression block at the current time. $r_{t}$ is set to 0 or 1, which means whether the entire historical information is used at the current time. The larger the value of R, the more information available at the previous time. The specific calculation formula is shown in (1):(1)$$r_{t}=\sigma \left(W_{rx}x_{t}+W_{rh}h_{t-1}+b_{r}\right),$$
where $x_{t}$ is the input at time t, $W_{rx}$ is the weight of $x_{t}$, $W_{rh}$ is the weight of $h_{t-1}$, $b_{r}$ is the offset of the reset gate $r_{t}$, and σ(•) is the sigmoid function. The update gate $z_{t}$ is used to control the percentage of historical information used during modeling. Similar to the reset gate $r_{t}$, the larger the value of the update gate, the larger the amount of historical information of the loop block used. The calculation formula is shown in Formula (2):(2)$$z_{t}=\sigma \left(W_{zx}x_{t}+W_{zh}h_{t-1}+b_{z}\right),$$
where $W_{zx}$ is the weight of $x_{t}$, $W_{zh}$ is the weight of $h_{t-1}$, and $b_{\tilde{h}}$ is the offset of update gate.

The calculation formula of the hidden candidate state ${\tilde{h}}_{t}$ is as follows (3):(3)$${\tilde{h}}_{t}=\tanh\left(W_{\tilde{h}t}x_{t}+W_{\tilde{h}h}\left(r_{t}⨀h_{t-1}\right)+b_{h}\right),$$
where $W_{\tilde{h}x}$ is the weight of $x_{t}$, $W_{\tilde{h}h}$ is the weight of $h_{t-1}$, and $b_{\tilde{h}}$ is the offset of hidden candidate states. ⨀ denotes element multiplication.

The calculation formula of the hidden state $h_{t}$ at the current time is as follows (4):(4)$$h_{t}=\left(1-z_{t}\right)⨀h_{t-1}+z_{t}⨀{\tilde{h}}_{t},$$

Due to the excellent characteristics of GRU [25], GRU is selected as the selection network in this paper. Additionally, three different memory numbers are used to achieve smooth feature extraction in descending order. Dropout is used between GRU layers to prevent overfitting.

### 2.2. Pearson Correlation Coefficient

The diagnosis accuracy of fault diagnosis using neural network depends on the selected data and features to a large extent. Artificial intelligence methods combined with discriminative features can improve fault diagnosis accuracy. To extract features that contain more fault information, we use Pearson correlation coefficients combined with fault states to select sensors and their time domain features.

Pearson correlation coefficient is widely used to measure the degree of correlation between two variables. To obtain low data redundancy and distinguish the relationship between different wear quantities to the greatest extent, Pearson correlation coefficient is used to select the signal and feature with the largest fault information content [26,27]. The data of the two sensors A and B are $A=\left(a_{1},a_{2}\cdots a_{n}\right)$ and $B\left(b_{1},b_{2}\cdots b_{n}\right)$, respectively, and N represents the length of the data segment. Then, the Pearson correlation coefficient $\rho_{A,B}$ between the two groups of data is calculated as follows (5):(5)$$\rho_{A,B}=\frac{\mathrm{cov}\left(A,B\right)}{\sigma_{A}\sigma_{B}}=\frac{E\left[\left(A-\mu_{A}\right)\left(B-\mu_{B}\right)\right]}{\sigma_{A}\sigma_{B}},$$

### 2.3. Methods

#### 2.3.1. Construction of Test Bench

To verify the effectiveness of the proposed method, we collected signals using a fault simulation test bench for experimental validation. The high-speed engine fault simulation test bench is composed of a diesel engine, transmission shaft, DC motor, console and relevant accessories. The schematic diagram is shown in Figure 2.

A certain type of diesel engine is selected for the test. It adopts ECU (Electronic control unit) and electronic control high-pressure common rail technology. It is a four-stroke six-cylinder, water-cooled, in-line, and dry-cylinder sleeve type.

The signals monitored in the test include top dead center, cylinder head surface vibration, fuselage surface vibration, fuselage bottom vibration, and cylinder pressure signal. Wherein the cylinder pressure sensor is installed at the position of cylinder 6. To make the measured vibration signal reflect the cylinder pressure excitation as much as possible and reduce the interference of other factors, we installed a vibration acceleration sensor above the cylinder head to measure the vibration signal. The vibration signal measuring point of the fuselage is set in the middle of the fuselage surface.

In the test, the no-load speed of the diesel engine is 2100 R/min. The working state of the cylinder liner of the diesel engine under four different wear amounts is simulated. The sampling frequency and sampling time are 40,960 Hz and 0.8 s.

The description of multiple fault diagnosis data sets is shown in Table 1.

#### 2.3.2. Selection Features of Double Pearson Correlation Coefficient

In the paper, we use Pearson correlation coefficients for the selection of time domain features of vibration signals.

A.Sensor signal selection

To select the vibration signal containing more fault information, we used the Pearson correlation coefficient combined with the fault state for the vibration signal selection. For a certain sensor, such as the cylinder head vibration signal, we calculate the Pearson correlation coefficient of the cylinder head vibration signal for each of the two fault states for all fault states. As the absolute value of the Pearson correlation coefficient is close to 0, it means that the data are more irrelevant, i.e., it is easier to distinguish between different fault states. We calculate the absolute value of the Pearson correlation coefficient for different sensors combined with the fault state, compare them, and select the sensor with the smallest absolute value as the sensor selected in this paper.

Since the top dead center signal and the cylinder pressure signal are slowly changing signals and contain less fault information, the top dead center signal and the cylinder pressure signal are dropped. Take the cylinder fault severity as an example, calculate the Pearson correlation coefficient of the cylinder pressure signal in four fault states by using Equation (5), and then calculate the Pearson correlation coefficient of the remaining three sensor signals in sequence. The cylinder diameters 126.20 mm, 126.40 mm, 126.60 mm, and 126.80 mm correspond to labels 2-1, 2-2, 2-3, and 2-4. The cylinder fault data set are shown in Table 2.

It can be seen from Figure 3 that the sum of absolute values of Pearson correlation coefficients of No. 1 sensor (cylinder head surface vibration signal) is the lowest, which indicates that the correlation degree is the lowest. So, the cylinder head surface vibration signal has the greatest discrimination degree under different fault states. Since there are four wear degrees of the cylinder and only three wear degrees of the other three fault types, it is the most difficult to diagnose the fault severity of the cylinder. Therefore, cylinder head surface vibration signal is selected for all single-fault diagnoses.

B.Feature selection

To extract time domain features containing more fault information from the sensor data, we performed the selection of time domain features using Pearson correlation coefficients combined with fault states. We extract 18 time domain features from the sensor selected in the above step. For a certain time domain feature, we calculate the Pearson correlation coefficients of the time domain feature for each two fault states for all fault states. As the absolute value of Pearson’s correlation coefficient is closer to 0, it means that the data are more irrelevant, i.e., it is also easier to distinguish between different fault states. Therefore, we compare the sum of the absolute values of Pearson correlation coefficients calculated from different time domain features combined with fault states and select the time domain feature with the smallest sum of absolute values as the time domain feature selected in this paper.

After the above calculation, we select the cylinder head surface vibration signal for the next step of feature extraction. Then we extracted 18 time domain features (mean value, peak value, root mean square, peak–peak value, average amplitude, mean square, root mean square, variance, standard deviation, waveform factor, peak factor, skewness, kurtosis, kurtosis factor, pulse factor, margin factor, skewness factor, and coefficient of variation, serial number 1–18) with a non-overlapping sliding window with a window length of 32. The formula is shown in Table 3. The Pearson correlation coefficients of 4 fault states of 18 time domain features are calculated in sequence, and it can be seen that the most irrelevant one is No. 18 (coefficient of variation). Figure 4 is a histogram of the sum of the absolute values of Pearson correlation coefficient of the number of four fault states of 18 time domain features.

As shown in Figure 4, the sum of the absolute values of the Pearson correlation coefficients of the No. 18 feature (coefficient of variation) is the lowest, which indicates that the coefficient of variation of different fault states has the largest degree of discrimination and is suitable for fault diagnosis. The coefficient of variation is finally selected as the final feature in this paper.

#### 2.3.3. Data Enhancement and Division

The single-fault diagnosis model is trained by using the coefficient of variation of the cylinder head surface vibration signal selected above as a feature. Since multi-fault diagnosis is more difficult and requires more state information than single-fault diagnosis, multi-fault diagnosis selects the features selected above from the three vibration signals for multi-fault diagnosis. Then, we obtain the data set for model training by data augmentation of the coefficient of variation with a sliding window method with a window length of 128 and a step size of 4. For single-fault diagnosis, the data set is divided into training and validation sets in a ratio of 8 to 2. We select datasets not used for model training and validation within the sliding window interval of the sliding window method as the test set to verify the generalization ability of the model. For multi-fault diagnosis, 20% of the data set is used as the test set, and the rest of the data is divided into training and validation sets in a ratio of 8 to 2.

#### 2.3.4. Model Training

In this section, the training of single-fault diagnosis model and multi-fault diagnosis model is performed.

The fault diagnosis performance of the proposed method is verified by experimental data. Computer hardware configuration: the processor is Intel (R) core (TM) i5-7300hq, the GPU graphics card is NVIDIA GeForce GTX 1050, the computer memory is 8 GB, the operating system is Windows10 (64 bit), the programming language is Python (version 3.8.8), the software framework structure is Keras deep learning tool, the Tensorflow deep learning framework is used as the back-end support, and the development software is Spyder.

In this paper, the three-layer GRU is selected as the selected network. The cross-entropy function for multi-classification is selected as the loss function. Additionally, the Adam optimizer with fast convergence speed and easy parameter adjustment is selected as the optimizer. The number of output neurons of the last fully connected layer is set according to different fault types (for example, the number of neurons of the fully connected layer of the other three fault types except cylinder fault is 3 (three fault severity)). In the single-fault diagnosis model, the dropout rate of only the cylinder fault diagnosis model is 0.5, and the dropout rate of the other three fault states is 0.2.

A.Single-fault diagnosis model training

Single-fault diagnosis is to diagnose the fault severity of a single fault type of the engine. All single-fault diagnostic models use the coefficient of variation of the cylinder head surface vibration signal extracted above as input for fault diagnosis. We train different fault diagnosis models for different fault types to carry out corresponding fault diagnoses. A total of four fault diagnosis models are trained.

The detailed parameter settings of single-fault diagnosis network model and multi-fault diagnosis network model are shown in Table 4, and the structure of single-fault (cylinder) diagnosis network is shown in Figure 5. The coefficient of variation of the cylinder head surface vibration signal selected in Section 2 is used as input for single-fault diagnosis model.

B.Multi-fault diagnosis model training

Multi-fault diagnosis is to diagnose the severity of multiple fault types of the engine. We train a multi-fault diagnosis model for the data of multiple fault types to obtain different fault severity results of multiple fault types.

The structural parameters of multi-fault diagnosis network are shown in Table 5 and the structural diagram of the multi-fault diagnosis network is as follows. Since multi-fault diagnosis is more difficult, the multi-fault diagnosis model uses the coefficients of variation of three vibration signals as input for fault diagnosis. The coefficient of variation of the three-dimensional vibration signal is stacked into 3 × 128 data and input into the multi-fault diagnosis model for multi-fault diagnosis. The multi-fault diagnosis network structure is shown in Figure 6.

## 3. Results

In this section, the single-fault diagnosis of four engine faults is carried out, and then the multi-fault diagnosis is carried out to directly obtain the fault severity of the corresponding fault type. To reduce the influence of random factors, five random experiments are carried out for each experiment, and the final experimental results are the average of the five random experiments.

### 3.1. Single-Fault Diagnosis

Four fault severity diagnosis models were trained for the four fault types. The effective features of four fault types (exhaust valve, cylinder, piston ring, and intake valve) are extracted and enhanced to obtain model input. Then, the training data set of four fault types are sent to four network models designed in this paper for training to obtain four single-fault diagnosis models.

Figure 7 are t-SNE dimension reduction diagrams of the 32-dimensional fully connected layer eigenvector of the single-fault diagnosis test set of four faults (exhaust valve, cylinder, piston ring, and intake valve). From the above feature reduction diagram, we can see that after combining the fault state with Pearson correlation coefficient and GRU, we can extract features with high resolution and distinguish between different fault severity. There are obvious differences between different fault severity, and the diagnosis results are obvious. The fault diagnosis method in this paper is effective.

In the four single-fault diagnosis models, the average test accuracy dimension of only five rounds of cylinders is 99.99%. The average test accuracy of the other three fault states reached 100%, and the generalization ability of the selected features was verified.

### 3.2. Multi-Fault Diagnosis

By training a multi-fault diagnosis model with the data of multiple fault types, different fault severity results of multiple fault types can be obtained. A dataset of 13 fault severities of four fault types (three wear degrees of the exhaust valve, four wear degrees of the cylinder, three wear degrees of the piston ring, and three wear degrees of the intake valve) is extracted by features to obtain the input of the multi-fault diagnosis model, which is then fed into the multi-fault diagnosis model for training.

The dimension reduction results of t-SNE before the full connection classification layer of multi-fault diagnosis are shown in Figure 8.

It can be seen from the feature reduction diagram that the difference between different severity of different faults is obvious, the extracted features have high expressive power, and there is no crossover between different fault severity features. The classification result is excellent.

Figure 9 is the confusion matrix of diagnosis results of the test set of five rounds of random experiments with thirteen faults.

From the confusion matrix of five rounds of random experiments, the average test accuracy of five rounds of random experiments can reach 99.97%, and the generalization ability and diagnostic accuracy are excellent.

### 3.3. Analysis of Influencing Factors

#### 3.3.1. Dropout Rate

To prevent overfitting, dropout is used in the network. However, a too large dropout rate will lead to underfitting, and a too small dropout rate will lead to overfitting. Therefore, choosing a suitable dropout rate is very important for the performance of the model. Five rounds of crossover experiments were carried out with the dropout rates of 0.2, 0.3, 0.4, and 0.5. The results of five rounds of crossover experiments with different dropout rates are shown in Table 6.

According to the results in Table 6, when the dropout rate is 0.2, the average test accuracy and average verification accuracy of the multi-fault model are the highest, and the results verify the superiority of the dropout rate selected by the model.

#### 3.3.2. Number of Vibration Signals Selected

To verify the necessity of selecting the features of the three vibration signals as inputs, this subsection only uses one and two sensor signals (1 × 128 and 2 × 128) as inputs to the network for fault diagnosis to verify the necessity of the three sensor signals for multi-fault diagnosis.

To avoid the influence of random factors, the final results are taken as the average of five rounds of random experiments. The results of five rounds of crossover experiments with different numbers of vibration signals are shown in Table 7.

From the results in Table 7, we can conclude that the average test accuracy and the average validation accuracy of the multi-fault diagnosis model are the highest when the number of sensors is 3. Therefore, it is necessary to use the three sensor signals for multi-fault diagnosis.

### 3.4. Comparison of Different Feature Extraction Methods

To verify the effectiveness of the proposed feature extraction method, two feature extraction methods, PCA (principal component analysis) and CNN, are used to extract features from vibration signals. The extracted features are input to GRU for multi-fault diagnosis.

To avoid the influence of random factors, the final results are taken as the average of five rounds of random experiments. The results of five rounds of crossover experiments with different feature extraction methods are shown in Table 8.

From the results in Table 8, we can conclude that the average test accuracy and average verification accuracy of the proposed feature extraction method in the paper are the highest. The effectiveness of the proposed method is verified.

### 3.5. Comparison of Different Networks

To verify the effectiveness of the network model (GRU) used, two network models, LSTM and RNN, were used for comparison.

To verify the effectiveness of the proposed feature extraction method, two feature extraction methods, PCA (principal component analysis) and CNN, are used to extract features from vibration signals. The extracted features are input to GRU for multi-fault diagnosis.

To avoid the influence of random factors, the final results are taken as the average of five rounds of random experiments. The results of five rounds of crossover experiments with different networks are shown in Table 9.

From the results in Table 9, we can conclude that the network used in the article, GRU, has the highest average test accuracy and average verification accuracy. The network used in the paper is reasonable.

## 4. Conclusions

As the power source of important equipment, the normal operation of the engine is very important to ensure the safety of life and property. Therefore, we must use reasonable engine fault diagnosis [28,29] technology to diagnose the engine status.

Most of the previous methods only diagnose the fault type of the engine without further analyzing the fault severity of the engine fault type, which is not conducive to taking reasonable maintenance measures to ensure the safety of life and property.

To select the sensor signal with the most fault information and the time domain features, the variation coefficient of the cylinder head surface vibration signal is selected as the feature by using the double Pearson correlation coefficient and combined with GRU for fault diagnosis. The experimental results of single-fault diagnosis prove that the double Pearson coefficient can extract effective features. At the same time, the complexity of multi-fault diagnosis is higher than that of single-fault diagnosis, and more fault state information is required. Therefore, by stacking the variation coefficients of three vibration signals as inputs and sending them to the fault diagnosis network for fault diagnosis, the fault type and severity of the engine can be accurately diagnosed. The effectiveness of this method is verified, and the average test accuracy of the test set reaches 99.97%. The generalization ability of the method is also verified.

## Figures and Tables

**Figure 1 sensors-22-08164-f001:**
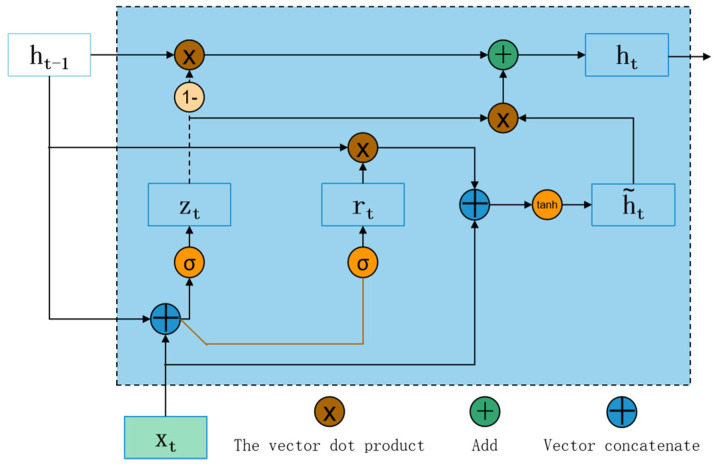
GRU unfolding structure.

**Figure 2 sensors-22-08164-f002:**
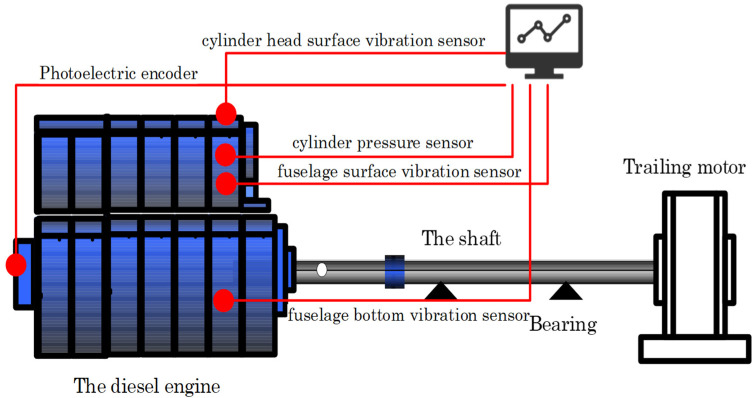
The schematic diagram.

**Figure 3 sensors-22-08164-f003:**
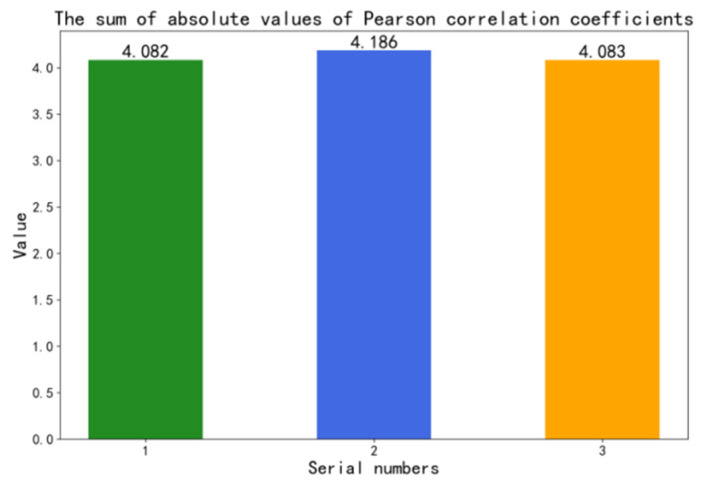
The sum of absolute values of Pearson’s correlation coefficients.

**Figure 4 sensors-22-08164-f004:**
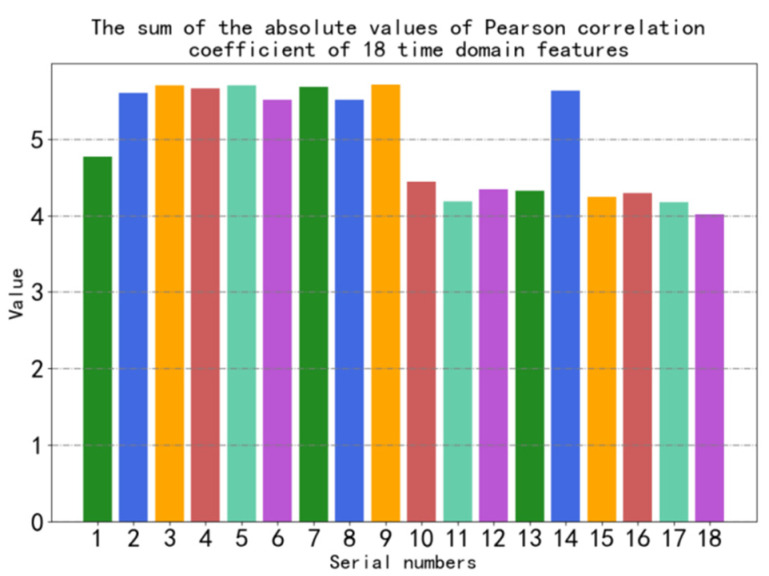
The sum of the absolute values of Pearson correlation coefficient of 18 time domain features.

**Figure 5 sensors-22-08164-f005:**
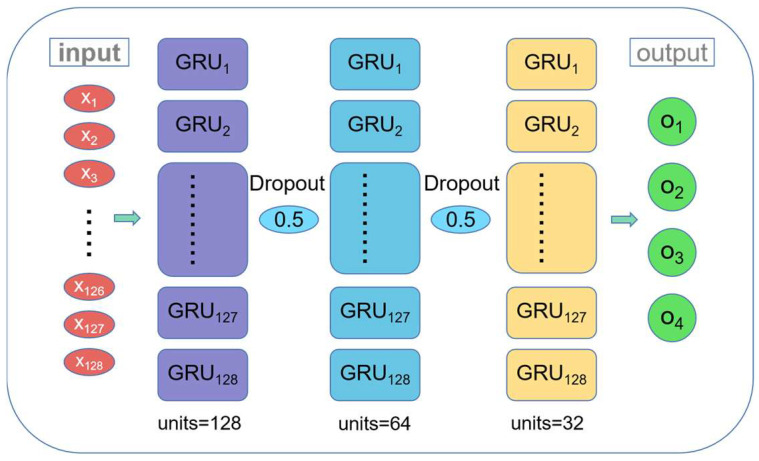
The structure of single-fault (cylinder) diagnosis network.

**Figure 6 sensors-22-08164-f006:**
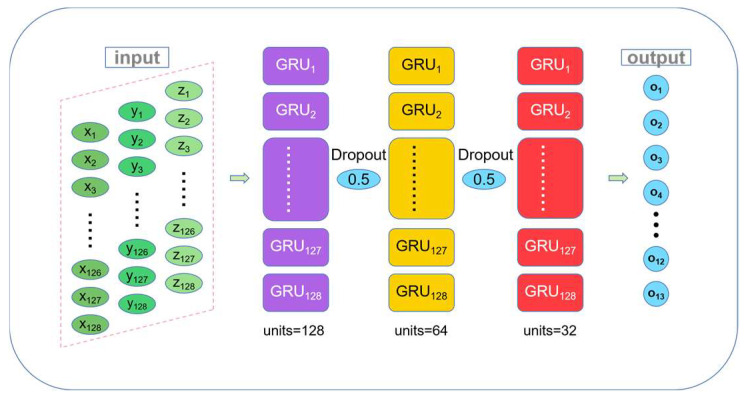
The multi-fault diagnosis network structure.

**Figure 7 sensors-22-08164-f007:**
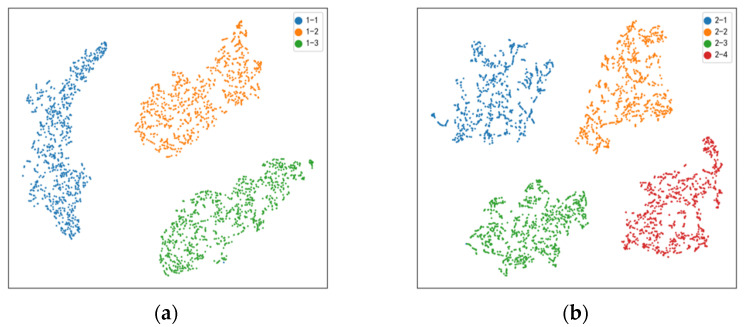
The t-SNE dimension reduction diagram of four faults ((**a**–**d**) exhaust valve, cylinder, piston ring, and intake valve).

**Figure 8 sensors-22-08164-f008:**
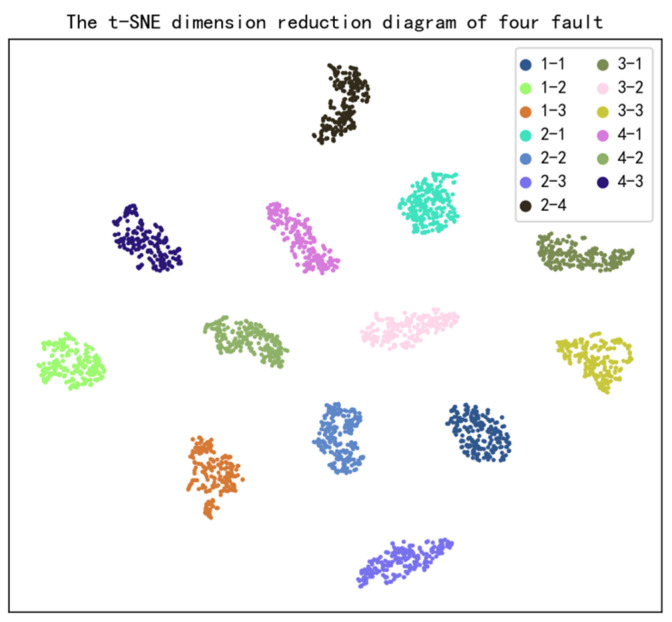
T-SNE dimension reduction diagram of multi-fault diagnosis network features.

**Figure 9 sensors-22-08164-f009:**
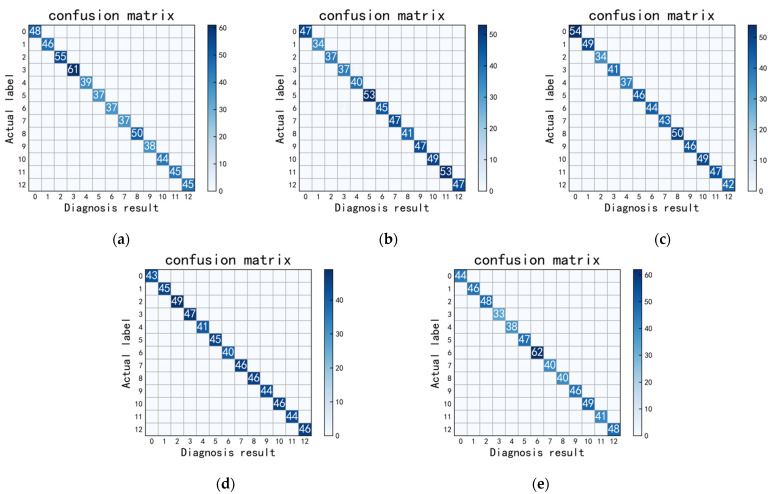
Five rounds of random experiments with thirteen faults ((**a**–**e**) First to fifth rounds of random experiments).

**Table 1 sensors-22-08164-t001:** Multi-fault diagnosis data set.

Fault Type Label	Fault Type	Fault Severity	Fault Severity Label
1	Exhaust valve wear	0.3 mm	1-1
		0.7 mm	1-2
		1.0 mm	1-3
2	Cylinder wear	126.20 mm	2-1
		126.40 mm	2-2
		126.60 mm	2-3
		126.80 mm	2-4
3	Piston ring wear	1.0 mm	3-1
		2.0 mm	3-2
		3.0 mm	3-3
4	Intake valve wear	0.1 mm	4-1
		0.6 mm	4-2
		0.9 mm	4-3

**Table 2 sensors-22-08164-t002:** Cylinder fault data set.

Sensor Signal	Fault Label	2-1	2-2	2-3	2-4
Cylinder head surface vibration signal	2-1	1	−0.147432	−0.054711	−0.157429
	2-2	−0.147432	1	0.06385	−0.137345
	2-3	−0.054711	0.06385	1	−0.147416
	2-4	−0.157429	−0.137345	−0.147416	1
Fuselage surface vibration signal	2-1	1	−0.015744	−0.001972	0.013253
	2-2	−0.015744	1	0.003264	0.006259
	2-3	−0.001972	0.003264	1	−0.001006
	2-4	0.013253	0.006259	−0.001006	1
Fuselage bottom vibration signal	2-1	1	−0.005371	−0.018656	0.008554
	2-2	−0.005371	1	0.001913	−0.046587
	2-3	−0.018656	0.001913	1	−0.012417
	2-4	0.008554	−0.046587	−0.012417	1

**Table 3 sensors-22-08164-t003:** Time domain features.

Name	Formula	Name	Formula
Mean value	$\bar{X}=\frac{1}{N}\sum_{i=1}^{N}x_{i}$	Waveform factor	$S=\frac{X_{\mathrm{rms}}}{\left|\bar{X}\right|}$
Peak value	$X_{p}=\max\left\{\left|x_{i}\right|\right\}$	Peak factor	$C=\frac{X_{p}}{X_{\mathrm{rms}}}$
Root mean square	$X_{\mathrm{rms}}=\sqrt{\frac{1}{N}{\sum_{i=1}^{N}x^{2}{}_{i}}^{}}$	Skewness	$S=\frac{1}{N}{\sum_{i=1}^{N}\left(\frac{x_{i}-\bar{X}}{\sigma }\right)}^{3}$
Peak–peak value	$X_{p-p}=\max\left\{x_{i}\right\}-\min\left\{x_{i}\right\}$	Kurtosis	$K=\frac{\beta }{X^{4}{{}_{rms}}^{}}$
Average amplitude	$\left|\bar{X}\right|=\frac{1}{N}\sum_{i=1}^{N}\left|x_{i}\right|$	Kurtosis factor	$\beta =\frac{1}{N}\sum_{i=1}^{N}x^{4}{}_{i}$
Mean square	${\bar{X}}^{2}=\frac{1}{N}\sum_{i=1}^{N}x^{2}{}_{i}$	Pulse factor	$I=\frac{X_{P}}{\bar{X}}$
Root mean square	$X_{r}={\left(\frac{1}{N}\sum_{i=1}^{N}\sqrt{\left|x_{i}\right|}\right)}^{2}$	Margin factor	$L=\frac{X_{P}}{X_{r}}$
Variance	$\sigma^{2}=\frac{1}{N-1}\sum_{i=1}^{N}{\left(x_{i}-\bar{x}\right)}^{2}$	Skewness factor	$IPD=\frac{X_{P}}{X^{3}{}_{\mathrm{rms}}}$
Standard deviation	$\sigma^{}=\sqrt{\frac{1}{N-1}\sum_{i=1}^{N}{\left(x_{i}-\bar{x}\right)}^{2}}$	Coefficient of variation	$C_{v}=\frac{\sigma }{{\bar{X}}_{}}$

**Table 4 sensors-22-08164-t004:** Structural parameters of single-fault (cylinder) diagnosis network.

Structure	Input	Output	Memory Numbers	Learning Rate	Activation Function
Input	(128, 1)	(128, 1)			
GRU	(128, 1)	(128, 128)	128		Tanh
Dropout	(128, 128)	(128, 128)		0.5	
GRU	(128, 128)	(128, 64)	64		Tanh
Dropout	(128, 64)	(128, 64)		0.5	
GRU	(128, 64)	(32, 1)	32		Tanh
Dense	(32, 1)	(4, 1)			Softmax

**Table 5 sensors-22-08164-t005:** The structural parameters of the multi-fault diagnosis network.

Structure	Input	Output	Memory Numbers	Learning Rate	Activation Function
Input	(128, 3)	(128, 3)			
GRU	(128, 3)	(128, 128)	128		Tanh
Dropout	(128, 128)	(128, 128)		0.5	
GRU	(128, 128)	(128, 64)	64		Tanh
Dropout	(128, 64)	(128, 64)		0.5	
GRU	(128, 64)	(32, 1)	32		Tanh
Dense	(32, 1)	(13, 1)			Softmax

**Table 6 sensors-22-08164-t006:** Results of five round crossover experiments with different dropout rates.

Dropout Rate	Average Test Accuracy	Average Verification Accuracy
0.2	99.97%	100.00%
0.3	98.35%	99.86%
0.4	98.69%	100.00%
0.5	99.86%	100.00%

**Table 7 sensors-22-08164-t007:** Results of five round crossover experiments with different numbers of vibration signals.

Number	Average Test Accuracy	Average Verification Accuracy
1	99.45%	99.62%
2	99.18%	99.74%
3	99.97%	100.00%

**Table 8 sensors-22-08164-t008:** The results of five rounds of crossover experiments with different feature extraction methods.

Methods	Average Test Accuracy	Average Verification Accuracy
PCA	53.36%	53.65%
CNN	51.31%	59.44%
The proposed method	99.97%	100.00%

**Table 9 sensors-22-08164-t009:** The results of five rounds of crossover experiments with different networks.

Networks	Average Test Accuracy	Average Verification Accuracy
LSTM	90.21%	95.06%
RNN	18.59%	27.51%
GRU	99.97%	100.00%

## Data Availability

Not applicable.

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
