# Peer review of "Research on Multi-Fault Diagnosis Method Based on Time Domain Features of Vibration Signals"

_sensors, 2022, doi:10.3390/s22218164_

Round 1

Reviewer 1 Report

the paper applied Pearson correlation coefficient and GRU method into fault diagnosis. the fault severity type and severity can be diagnosed through the time-domain feature stacking of multi-dimensional vibration signals. The results show the effectiveness of the method and the selected model parameters. some comments:

1)      the title shows the main contribution of the paper focus on time series characteristic analysis. But the author only applied common time domain method in to model. What is the main innovation for the method? How about the author improve Pearson correlation coefficient analysis method sucha as self-adaption based on the characteristic of vibration signal.

2)      The author shows the method can diagnose the severity of the fault. As we know, the author only train the data with different severity fault. And based on the training model, the fault of different severity can be diagnosed. It is still the same with the fault classification.

Reviewer 2 Report

In this paper, the authors propose a ‘Research on multi fault diagnosis method based on time domain characteristics of vibration signals.’ In general, the research point of this paper meets the scope of Sensor. Here is my specific suggestion:

(1)    The Abstract of this paper should be rewritten such as “the maintenance of the engine fault status is of great significance for ensuring the safety of life and property”. It seems wrong in the expression of this sentence. The aim of our research is the normal operation of the machine which is not what you called the maintenance of the engine fault status. Besides, the gating Recurrent unit should be written in Gating Recurrent Unit.

(2)    Many words in this paper should be written in the full name such as SVM, CNN, DBN, ECU, etc.

(3)    The language of this paper should be polished. Many expressions in this paper exist the problems and the logicality of this paper should be improved. Such as the sentence in lines 141-142, the logicality of the sentence is confusing. What is the number of figures in line 152?

(4)    What is the core innovation of this paper? Is that the neural network i.e., GRU or Pearson correlation coefficient? Although the author finds out the limitation of related methods and solves the problems. But, the innovation of this paper is limited.

(5)    The quality of Figure 2 should be improved. Besides, the comparative result is not obvious in Figure 3. The comparative result in the figure can be partially enlarged.

(6)    Where is the comparison with the different methods? In the comparison of this paper, the different dropouts are used for analysis. Actually, this step only is the parameter selection in the training phase.

(7)    In section 3.3.2, the diagnosis accuracy of the 3 vibration signals method can be improved compared with the other methods. However, the computing complexity should also be considered in your proposed method.

(8)    It is suggested that some sentences of the conclusion can be moved to the introduction to further illustrate the priority of the proposed method. After careful reading of this paper, the core innovation of this paper relies on sensor signal selection and feature selection.

Reviewer 3 Report

The paper is interesting but needs serious revision relative to English and phrase building. 

Please consider the following remarks when making correctioins:

English must be seriosly revised.

Only few examples of detected problems:

(a)Phrase from lines 23...26 needs revision.

(b) More phrases , (not one!) are at lines 61...67, but some punctuation marks/words are either missing, or are used incorrectly.

(c) In many lines , comma is used instead of dot and this generated confusions.

(d) „;” should not be used after a title

Formal problems:

(a)    Abbreviation should be explained at their first mention in the paper (e.g. VMD, BP etc.)

(b)    The first line after an equation used to explain the significance of terms should not be aligned with a TAB and should start with a low case letter („where...” instead of „Where”)

(c)     Lie 224 – it is better to use „A” instead of „1” (see also line 297).  Line 246 – it is better to use „B” instead of „2” (see also line 309)

(d)    Lines 284...288 – first letter should be uppercase for python, tensorflow etc.

(e)     Figures 5 and 6 have illegible (too small) texts.

(f)     References disobey the template, different fonts are used and some words are truncated to the right edge.

Content problems:

(a)I do not think that the word „materials” represents a proper selection for the titles of section 2 and subsection 2.1

(b) Line 232: label 0 is mentioned whilst label 4 is missing (Table 1 refers to labels 1...4!)

Round 2

Reviewer 1 Report

I am afraind in can not agree the paper for publication by current version.

i have read the reply. i am sorry the authors did not revise the paper. thay only give me some explaination such as they wil do it in future

Reviewer 2 Report

Many of my concerns had been addressed properly. However, the expression of this paper remains exists some problems. Such as the sentence in lines 55-59. The logic of the sentence is confusing. Why do these two sentences repeat the same logical connective? This paper can be accepted in the current version.
